# Aromatic oil from lavender as an atopic dermatitis suppressant

**Haruna Sato[1], Kosuke Kato[1], Mayuko Koreishi[1], Yoshimasa Nakamura[2], Yoshio Tsujino[3], Ayano Satoh[1]***

**1** Graduate School of Interdisciplinary Science and Engineering in Health Systems, Okayama University, Okayama, Japan, **2** Graduate School of Environmental and Life Science, Okayama University, Okayama, Japan, **3** Graduate School of Science, Technology, and Innovation, Kobe University, Kobe, Hyogo, Japan

* ayano113@cc.okayama-u.ac.jp

**Data Availability Statement:** All relevant data are within the manuscript and its Supporting Information files.

**Funding:** This work was supported in part by MEXT/JSPS KAKENHI Grant Numbers 18K06133

## Abstract

In atopic dermatitis (AD), nerves are abnormally stretched near the surface of the skin, making it sensitive to itching. Expression of neurotrophic factor Artemin (ARTN) involved in such nerve stretching is induced by the xenobiotic response (XRE) to air pollutants and UV radiation products. Therefore, AD can be monitored by the XRE response. Previously, we established a human keratinocyte cell line stably expressing a NanoLuc reporter gene downstream of XRE. We found that 6-formylindolo[3,2-b]carbazole (FICZ), a tryptophan metabolite and known inducer of the XRE, increased reporter and Artemin mRNA expression, indicating that FICZ-treated cells could be a model for AD. Lavender essential oil has been used in folk medicine to treat AD, but the scientific basis for its use is unclear. In the present study, we investigated the efficacy of lavender essential oil and its major components, linalyl acetate and linalool, to suppress AD and sensitize skin using the established AD model cell line, and keratinocyte and dendritic cell activation assays. Our results indicated that lavender essential oil from *L. angustifolia* and linalyl acetate exerted a strong AD inhibitory effect and almost no skin sensitization. Our model is useful in that it can circumvent the practice of using animal studies to evaluate AD medicines.

## Introduction

Atopic dermatitis (AD) is a skin disease characterized by itchy eczema that repeatedly exacerbates and remits. It affects 15%–20% of children and 1%–3% of adults worldwide and patient numbers are increasing [1]. Genetic and environmental factors are thought to be involved in AD development. Approximately 20% of AD patients have genetic mutations in filaggrin that is involved in the epidermal barrier function [2], which impairs the barrier function and facilitates antigen invasion. Antigen invasion leads to the production of inflammatory cytokines such as thymic stromal lymphopoietin and interleukin-4, which in turn triggers inflammatory reactions such as allergic reactions, leading to the development or exacerbation of AD. AD can also develop and be aggravated by excessive production of immunoglobulin E due to genetic mutations in interleukin-12β, involved in the production of interferon-γ, which suppresses immunoglobulin E production [3], resulting in an excessive immune response to antigen

and 22K06128 to A.S., and MEXT/JSPS KAKENHI Grant Numbers 17H03818 and 20H02933 to Y.N. The funders had no role in study design, data collection and analysis, decision to publish, or preparation of the manuscript.

**Competing interests:** The authors have declared that no competing interests exist.

invasion. Environmental factors, including antigens such as mites and pollen, trigger an inflammatory response and stress [4] that affect the nervous system and cause itching. Chemicals such as air pollutants are also involved in the onset and exacerbation of AD via aromatic hydrocarbon receptor (AhR) [5]. Although air pollution has improved in recent years in many countries, the increase in the variety of chemicals associated with modernization is thought to have increased daily exposure to chemicals and there is the concern that these factors may increase the incidence of AD. Steroids and immunosuppressive agents are used to treat AD. However, they are only symptomatic treatments, and AD patients in remission may experience reoccurrence with worsening symptoms due to antigen invasion. Recent advances have seen new drugs such as dupilumab, baricitinib and tralokinumab (an anti-IL-13 agent) approved by the Food and Drug Administration and the European Medicines Agency [6]. These drugs act on specific pathogenic mechanisms of AD and represent a significant step forward. However, it is important to emphasize that despite these recent developments in AD pharmacotherapy, there remains an urgent need for the discovery of novel therapeutic agents capable of more effectively controlling AD. Therefore, a curative agent is desired for AD. Interestingly, in folk medicine, lavender essential oil is used to treat AD [7].

Lavender essential oil is one of the most widely distributed essential oils worldwide and is used in perfumes and cosmetics. Lavenders used for essential oil include *Lavandula angustifolia*, *Lavandula spica*, and *Lavandula stoechas*. Each has different major components with *L. angustifolia* containing linalyl acetate and linalool, *L. spica* containing linalool, 1,8-cineole, camphor [8], and *L. stoechas* containing fenchone terpene and camphor [9], although the major components and their contents may vary depending on the region and harvest time. Lavenders with a low camphor content and high terpene content such as *L. angustifolia* are used in perfumes and cosmetics, whereas lavenders with a high camphor content, such as *L. spica* and *L. stoechas*, are used in insect repellents [10]. *L. angustifolia* essential oil is traditionally thought to have anti-inflammatory and wound-healing properties, which is used as a massage oil in aromatherapy [11]. For many years, the scientific basis for such use was unknown, but in recent years, in addition to its anti-inflammatory [12] and wound healing [13] effects, increasing evidence has supported the traditional use of *L. angustifolia* essential oil, albeit inconclusively, for its antibacterial [14] and analgesic [12] effects. *L. angustifolia* essential oil also has a history of use as a folk remedy for AD [7]. In fact, a blend of oils containing *L. angustifolia* essential oil has been reported to alleviate symptoms in NC/Nga mice, an AD model [15]. It has also been shown to have an inhibitory effect on psoriasis, a chronic inflammatory skin disease [9]. Thus, *L. angustifolia* essential oil is expected to have symptom-relieving effects on skin diseases.

Linalyl acetate and linalool are terpenes and the major components of *L. angustifolia* essential oil, accounting for 1.2%–59.4% and 9.3%–68.8% of the oil, respectively [16]. Both compounds have anti-inflammatory effects by inhibiting carrageenin-induced edema in rat models of inflammation [17] and inhibiting inflammatory cytokine production by macrophages exposed to lipopolysaccharides [11], and analgesic effects by inhibiting the pain response to formalin injection in rat models of acute inflammatory pain [18, 19].

Essential oils have long been used in folk medicine, but recent discoveries have expanded their potential applications. For example, neem oil, traditionally used for its medicinal properties, is now being explored for its bioactive compounds, which may have potential for the development of natural mosquito repellents [20]. In response to increasing consumer resistance to synthetic preservatives, essential oils have emerged as promising candidates for food preservation through encapsulation techniques and smart packaging systems, which have shown potential in protecting various foods from microbial spoilage, enhancing flavor and extending shelf life [21]. In addition, the chemical profile of essential oil from the Amazonian aromatic species *Croton campinarensis* Secco, A. Rosário & PE Berry has demonstrated the

complexity of essential oil safety and the need for continued research to fully understand their potential risks and benefits [22]. Taken together, these studies highlight the diverse applications of essential oils beyond traditional medicine, from pest control to food preservation, and underscore the need for continued research into their safe and effective use.

Chemicals such as air pollutants, which are environmental causes of AD, are involved in the development and exacerbation of AD via the intracellular receptor AhR [5]. AhR is a ligand-dependent transcription factor that normally complexes with heat shock protein 90 (Hsp90), hepatitis B virus X-associated protein [XAP2; also known as aryl hydrocarbon receptor-interacting protein (AIP)], and p23. It is localized in the cytoplasm [23], but when it binds to ligands such as air pollutants including dioxins, it translocates into the nucleus. In the nucleus, AhR dissociates from Hsp90, XAP2, and p23 to form a dimer with AhR hydrocarbon receptor nuclear translocator (ARNT). This dimer binds to the xenobiotic response element (XRE) and induces expression of the neurotrophic factor Artemin [5]. This induces a hyper itching state in which nerves normally located in the dermis are stretched to the epidermis, making the skin prone to itch at the slightest stimulus. As a result, skin is damaged by scratching and the barrier function is impaired, facilitating antigen entry and exacerbating AD symptoms. Therefore, induction of gene expression downstream of XRE by AhR (hereafter referred to as AhR activation) may be an indicator of AD. Importantly, AhR is activated by not only dioxins, but also various chemicals including food metabolites and components such as 6-formylindolo[3,2-b]carbazole (FICZ), indirubin, 2-(1′H-indole-3′-carbonyl)-thiazole-4-carboxylic acid methyl ester, indole-3-carbinol, catechins, and flavonoids [24].

Here, we found that *L. angustifolia* essential oil and its major component, linalyl acetate, exerted suppressive effects on AD using our AD cell model, a human epidermal keratinocyte-like cell line stably expressing the NanoLuc (NLuc) reporter gene under the control of an XRE (XRE-NLuc::HaCaT). This cell line can be treated with known AhR activator FICZ to induce reporter gene expression, thereby mimicking the AD-induced state induced by air pollutants. Our results were supported by downregulation of Artemin quantified by quantitative polymerase chain reaction (qPCR). Furthermore, the suppressive effects were likely due to AhR and ARNT protein degradation. Additionally, *L. angustifolia* essential oil and its major component linalyl acetate barely induced skin sensitization that can be a major concern for the agent use as an ointment.

## Materials and methods

All the chemicals used in this study were obtained from Nacalai tesque, Kyoto, Japan, otherwise specified.

### Cell culture

Human immortalized keratinocyte cell line HaCaT (#300493, CLI, Cosmo-bio, Tokyo, Japan), established from adult male skin, was maintained at 37˚C under 5%$CO_2$ in Dulbecco's Modified Eagle's Medium (DMEM) supplemented with 10% fetal bovine serum (FBS). When the cells reached about 80% confluence, the cells were washed with PBS and stripped with trypsin/EDTA, and suspended in a new medium. 10~20% of the cell suspension was added to a new dish with a medium. For the reporter assays, HaCaT cell stably expressing the pNL (NLucP/XRE/Hygro) Vector (#CS186808, Promega Japan, Tokyo, Japan) and pNL (NLucP/ARE/Hygro) Vector (#CS 180902, Promega), termed XRE-NLuc::HaCaT [25] and ARE-NLuc::HaCaT [26], respectively, were used. The maintenance of these cell lines was performed similarly to those for HaCaT cells. Human monocyte cell line U-937 (#9021, National Institutes of Biomedical Innovation, Health and Nutrition, Tokyo Japan), established from an adult male patient with histiocytic lymphoma, was maintained at a density of $1 \times 10^5 \sim 1 \times 10^6$ cells/mL in RPMI1640 supplemented 10%FBS.

## Reporter assay and cell proliferation assay

For the XRE reporter assay for AD evaluation, XRE-NLuc::HaCaT cells were plated in a 96-well white plate at 10,000 cells/well/100 μL and cultured. After 24 h, culture supernatants were removed, and the test chemicals diluted with the culture medium were added to cells and incubated for 23 h. Three microliters of WST-1 assay reagent (#MK400, Takara Bio, Shiga, Japan) were then added, and the cells were incubated for 1 h. The culture supernatants were transferred to a clear 96-well plate, and their absorbance at 490 nm was measured at 670 nm using an iMark microplate reader (Bio-Rad Japan, Tokyo, Japan) to estimate the cell viability. Ten microliters of the NLuc substrate (#N1120, Promega) were added to the cells remaining in the wells of the white plate. After 5 min of incubation, luminescence was measured using a GloMax Navigator Microplate Luminometer (Promega). The reporter expression measured as luminescence was corrected based on the number of viable cells obtained as the absorbance at 490 nm. Cell viability below 70% was considered toxic in this study. The test chemicals used in this study are listed in S1 Table in S1 File. The ARE reporter assay for the evaluation of the number two key event in the adverse outcome pathway (AOP) for skin sensitization was performed similarly to those for AD evaluation but used ARE-NLuc::HaCaT cells with 48-h incubations. Although this assay uses the reporter expressing cell line established in our laboratory [26], it is essentially compatible with the organization for economic co-operation and development (OECD) test guideline (TG) 442D [27], *in vitro* skin sensitization assay addressing keratinocyte activation on the adverse outcome pathway for skin sensitization, ARE-Nrf2 Luciferase Test Method. Fold induction, $EC_{1.5}$ and $CV_{70}$ were calculated using the equations 1~3 shown in the TG442D [27].

## cDNA preparation and qPCR

cDNA synthesis and qPCR were performed as described previously [28]. Briefly, the total RNA was extracted, and reverse transcribed using SuperPrep II Cell Lysis & RT Kit for qPCR (#SCQ-401, Toyobo, Tokyo, Japan), according to the manufacturer's instructions. The cDNA obtained was subjected to a quantitative polymerase chain reaction (qPCR) with THUNDER-BIRD Next SYBR qPCR Mix (#QPX-201, Toyobo) according to the manufacturer's protocol using the StepOne Real-Time PCR System (Thermo Fisher Japan, Tokyo, Japan). The expression of each gene obtained using qPCR was normalized to that of glyceraldehyde-3-phosphate dehydrogenase (GAPDH). The primer sets used in this study are listed in S2 Table in S1 File. The cycle threshold (Ct) value of GAPDH in each sample was used as an internal control.

## Sodium dodecyl sulfate–polyacrylamide gel electrophoresis and western blotting

Sodium dodecyl sulfate–polyacrylamide gel electrophoresis (SDS–PAGE) and western blotting were performed as described previously [29]. Cell lysates were prepared in lysis buffer [10 mM HEPES-KOH, pH 7.4, 100 mM KCl, 1 mM $MgCl_2$, 1% Triton X-100, and protease inhibitor cocktails (Nacalai)]. After incubation for 10 min on ice, the lysate was clarified by centrifugation at $14,000 \times g$ for 10 min. The supernatants were then subjected to SDS–PAGE on a 7.5% gel (Bio-Rad). The gel was electrotransferred to polyvinylidene fluoride or polyvinylidene difluoride membranes (pore size: 0.45 μm, Merck Group Japan, Tokyo, Japan). The proteins on the membrane were detected by incubation with diluted anti-AhR, anti-ARNT (Santa Cruz Biotechnology, Cosmo Bio Co., LTD, Tokyo, Japan), and anti-γ-tubulin (Sigma, Sigma-Aldrich Japan, Tokyo, Japan) followed by HRP-conjugated anti-mouse immunoglobulin G (Cell Signaling, CST Japan, Tokyo, Japan) using a charge-coupled device camera.

## Flow cytometry

Another *in vitro* skin sensitization assay, addressing the number three key event, activation of dendritic cells of the AOP for skin sensitization was performed based on the OECD TG442E [30]. U937 cells at a density of $1 \times 10^6$ cells/mL were seeded at 300 μL/well in 24 well plates with or without test compounds and incubated at 37˚C for 48 h under 5% $CO_2$. Cells were collected by centrifugation and resuspended in 300 μL of ice-cold 1%BSA containing PBS (FACS buffer). Cells in FACS buffer were stained with FITC mouse anti-human CD86 (#555657, Becton Dickinson (BD), Japan BD, Tokyo, Japan), and FITC mouse IgG1 kappa isotype control (#555748, BD) at 4˚C for 30 minutes while shading from light (tapping the tube 15 minutes after starting staining). Cells were washed twice with 100 μL of ice-cold FACS buffer and stained with propidium iodide (PI, #P3566, Thermo) at the final concentration of 3 μg/mL and measured on a FACS Calibur (BD). The live-cell population R2 was obtained by gating R1 from SSC/FSC cytograms and then by gating FL3 negative populations, i.e., viable cell population from R1. Cell viability and the Stimulation Index (S.I.) were calculated using the following equations, shown in TG442E [30]. $EC_{150}$ and $CV_{70}$ were calculated using the equations shown in the TG442E.

Eq. 1

Cell viability = (# of living cells (R2))/(# of total cells (R1)) × 100

Eq. 2

The Stimulation Index (S.I.) = ((% of CD86 positive treated cells)–(% of isotype control positive treated cells))/((% of CD86 positive control cells)–(% of isotype control positive control cells)) × 100

## Statistical analysis

Data for each experiment are presented as mean ± standard deviation (SD). Statistical significance was determined by a two-tailed t-test of independent 2 samples or one-way analysis of variance using Microsoft Excel for Mac Version 16. To assess the statistical significance of the effect of the independent variable on the dependent variable, we first performed Bartlett's test for homogeneity of variances. If the test resulted in a p-value below 0.05, we performed a one-way ANOVA to compare the means of the groups. Otherwise, we performed the Kruskal-Wallis test, which is a nonparametric alternative to ANOVA. We used a code written in Python and executed on the Google Colab platform to perform these tests, bartlett, kruskal, f_oneway in scipy.stats. In addition, to confirm the dose-dependency of the effect, we performed the Williams test, which is a post hoc test for comparing multiple independent groups with unequal variances. This test determines whether the differences between the means of the groups are significant and whether the effect is dose-dependent. We used code written in R (version 4.2.3) to perform the Williams test, Williams in nparcomp.

All statistical tests were performed at a significance level of 0.05. The results of the tests were interpreted based on the p-values obtained, with a p-value less than 0.05 indicating statistical significance.

## Results

### *L. angustifolia* essential oil and its major components exert an AD inhibitory effect by inhibiting AhR activation

*L. angustifolia* essential oil has an AD inhibitory effect, but the scientific basis is unclear. It is also unclear which chemicals in the essential oil are responsible for the inhibitory effect. We tested the AD inhibitory effect of *L. angustifolia* essential oil and its major component, linalyl

acetate, and linalool using our AD cell model. AhR activation in keratinocytes induces expression of neurotrophic factor Artemin under the control of an XRE, resulting in excess elongation of nerves to the epidermis and AD [5]. Our AD cell model mimics AD-induced conditions by adding known AhR activator FICZ to human keratinocyte-like cell line HaCaT stably expressing the NanoLuc luciferase reporter gene under the control of an XRE (XRE-N-Luc::HaCaT). By adding test substances to this model, the AD inhibitory effect of the test substance is evaluated by attenuation of reporter gene expression as an indicator. Using this model, we investigated whether *L. angustifolia* essential oil inhibited AhR activation, thereby exerting an AD inhibitory effect. The *L. angustifolia* essential oil used in this study was provided its composition analysis and composed mainly of linalyl acetate (43 wt%) and linalool (32 wt%), which was obtained from the Japan Medical Aromatherapy Association (Tokyo, Japan, S3 Table in S1 File). For comparison, the same test was conducted using generic lavender oil with an unknown type of lavender and components, hereafter called generic lavender oil. The reporter-expressing cell line XRE-NLuc::HaCaT was seeded and cultured for 24 h and then incubated for another 24 h in medium containing 0.5 µM FICZ and 0%–0.02% of each lavender essential oil. After adding the NLuc substrate, reporter gene expression was measured by a luminometer. Below we describe the inhibited AhR activation is AD inhibition, although the agents tested actually inhibit AhR activation, may not the AD itself, which is a very complex entity. Percentage inhibition of AD was calculated by reporter expression with FICZ alone as 0% indicating no inhibition and without FICZ as 100% indicating complete inhibition. Both *L. angustifolia* essential oil and generic lavender oil inhibited AD in a concentration-dependent manner with $IC_{50}$ values of 0.012% and 0.02%, respectively, indicating that the former exhibited a stronger inhibitory effect (Fig 1A). Because the *L. angustifolia* essential oil used here was composed mainly of linalyl acetate and linalool, we determined whether these two chemicals inhibited AD. Both linalyl acetate and linalool inhibited AD in a concentration-dependent manner with $IC_{50}$ values of 519 µM and >1000 µM, respectively, indicating that the former was more potent (Fig 1B). Assuming that this oil contains approximately 1/2 linalyl acetate, the $IC_{50}$ values of 0.012% and 519 µM are calculated to be roughly equivalent. Linalool has two unsaturated double bonds at two different locations, which can be easily oxidized [31]. 6,7-Dihydrolinalool (DHL) and tetrahydrolinalool (THL) are hydrogenated forms of linalool at one and two unsaturated double bond(s), respectively, which are used as flavoring agents. We also tested these two chemicals in our AD cell model. As shown in Fig 1B, all the tested chemicals inhibited in a concentration-dependent manner, and THL and DHL exhibited stronger inhibitory effects than linalool, but did as not strong as linalyl acetate.

### *L. angustifolia* essential oil and its major components inhibit induced Artemin expression

Because *L. angustifolia* essential oil and its major components exerted an AD inhibitory effect by inhibiting AhR activation, they should also inhibit the induction of Artemin expression under the control of the XRE. We next examined whether *L. angustifolia* essential oil and its major components linalyl acetate and linalool inhibited the induced Artemin expression by FICZ. HaCaT cells were seeded and cultured for 24 h and then treated them with 5 µM FICZ and the indicated amounts of essential oil and components, and then Artemin mRNA levels were determined by qPCR. *L. angustifolia* essential oil and generic lavender oil decreased FICZ-induced Artemin expression with $IC_{50}$ values of 0.00006% (Fig 2A). Linalyl acetate and linalool also decreased the induced Artemin expression with $IC_{50}$ values of 3.6 and 194 µM, respectively, indicating that linalyl acetate exerted a stronger inhibitory effect (Fig 2B). These results, together with those from the AD cell model shown in Figs 1 and 2, suggested that *L.*

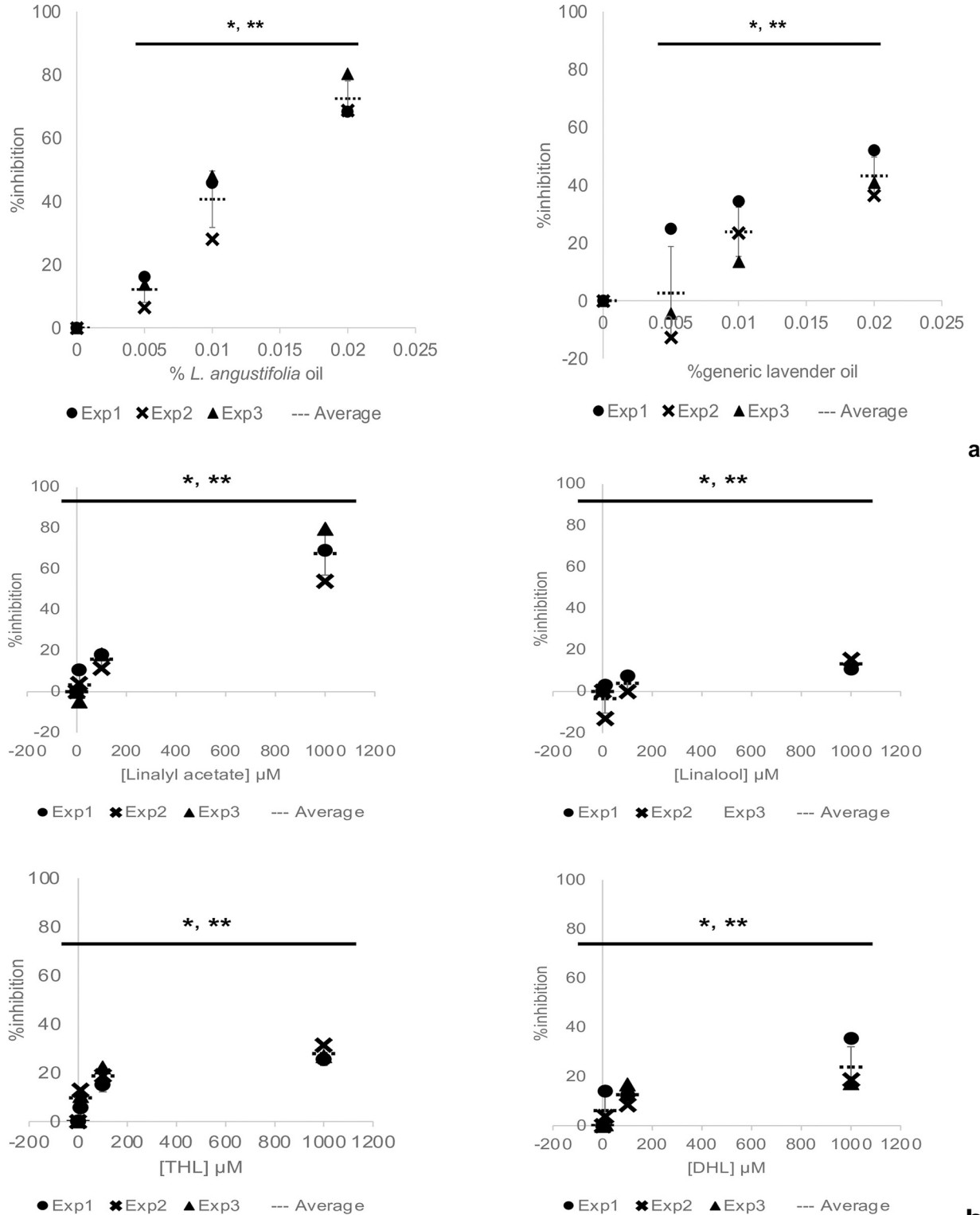

**Fig 1. *L. angustifolia* essential oil exerts an AD inhibitory effect by inhibiting AhR activation.** Human keratinocyte-like cell line HaCaT stably expressing the NanoLuc luciferase reporter gene under the control of an XRE (XRE-NLuc::HaCaT) was incubated in culture medium containing 0.5 μM FICZ with or without the indicated amounts of agents for 24 h. The percentage inhibition of AD was calculated by the reporter expression with FICZ alone as 0%, indicating no inhibition, and without FICZ as 100% indicating complete inhibition. The dot markers in the graph represent data from three independent experiments. The vertical and horizontal bars represent the standard deviation and the mean,

respectively. a) Comparison of *L. angustifolia* essential oil and generic lavender oil. b) Comparison of linalyl acetate, linalool, THL, and DHL. The statistical significance of the data points among the different concentrations and the dose-dependency were determined by the Kruskal-Wallis test or one-way ANOVA, and the Williams test, respectively, respectively. *, **$P<0.05$ (n = 3).

*angustifolia* essential oil, linalyl acetate, and linalool had an inhibitory effect on AD caused by chemicals such as air pollutants.

## Linalyl acetate may promote degradation of AhR and ARNT

We found that *L. angustifolia* essential oil and its major components, linalyl acetate and linalool, had an inhibitory effect on AD (Figs 1 and 2), although the underlying mechanisms were unclear. Without activation of AhR by FICZ, linalyl acetate decreased XRE reporter expression, suggesting that the inhibitory effect may not be due to the interaction between linalyl acetate and FICZ, but rather direct inhibition of AhR. To elucidate the mechanisms underlying this inhibitory effect, we first determined whether linalyl acetate affected nuclear translocation by observing GFP-fused AhR [32] and western blotting cellular fractions [33] with antibodies against AhR and ARNT whose binding to AhR is required for AhR transcriptional activity. However, unfortunately, we did not detect specific nuclear translocation by either method (S2 Fig in S1 File). Next, we measured the mRNA and protein expression of AhR and ARNT by qPCR and western blotting, respectively. As shown in Fig 3A and 3B, any treatment such as FICZ with or without linalyl acetate or linalool did not change the mRNA level of either AhR or ARNT, indicating that the inhibitory effect may not be due to downregulation of AhR or ARNT at the transcriptional level. Fig 3C shows the protein levels after each treatment and their quantitation is shown in Fig 3D. By FICZ and linalyl acetate treatments, the AhR protein level was reduced by approximately 30% compared with FICZ alone. Furthermore, the ARNT protein level, which was increased by FICZ alone, was decreased by FICZ and linalyl acetate treatments in a linalyl acetate concentration-dependent manner. These results suggest that linalyl acetate suppressed AD by promoting AhR and ARNT protein degradation under the tested conditions.

## *L. angustifolia* essential oil and its major components are positive in a keratinocyte activation assay for skin sensitization evaluation

To test the skin-sensitizing potential of *L. angustifolia* essential oil and its major components, we performed a keratinocyte activation assay that evaluates skin sensitization, the second key event out of four events defined in the AOP for skin sensitization by OECD TG442D [27]. We used keratinocyte-like cell line HaCaT stably expressing a luminescent reporter downstream of the antioxidant response sequence ARE (ARE-NLuc::HaCaT) established in our previous study [26]. The reporter-expressing cell line was incubated with test substances for 48 h and then induction of the reporter expression was measured. The positive criteria of this assay are 1) positive control is positive, 2) $EC_{1.5}$ is less than 1000 µM with more than 70% cell viability, and 3) clear overall dose-dependency of reporter induction. When the positive criteria are met in two or three independent tests, the test substance is considered positive. The $EC_{1.5}$ and $CV_{70}$ for each test substance are shown in Table 1. The positive control, ethylene glycol dimethacrylate (EGDMA), had an $EC_{1.5}$ of 4.1 ± 0.52 µM and a CV70 of 523 ± 59.0 µM. These met the positive criteria, indicating that EGDMA was positive. The reported $EC_{1.5}$ values range from 5 to 125 µM [34]. Although outside this range, the values obtained in this study were close, Therefore, the test was considered valid. The assay was conducted to test *L. angustifolia* essential oil and generic lavender oil with $EC_{1.5}$ values of 0.0072% ± 0.00065% and

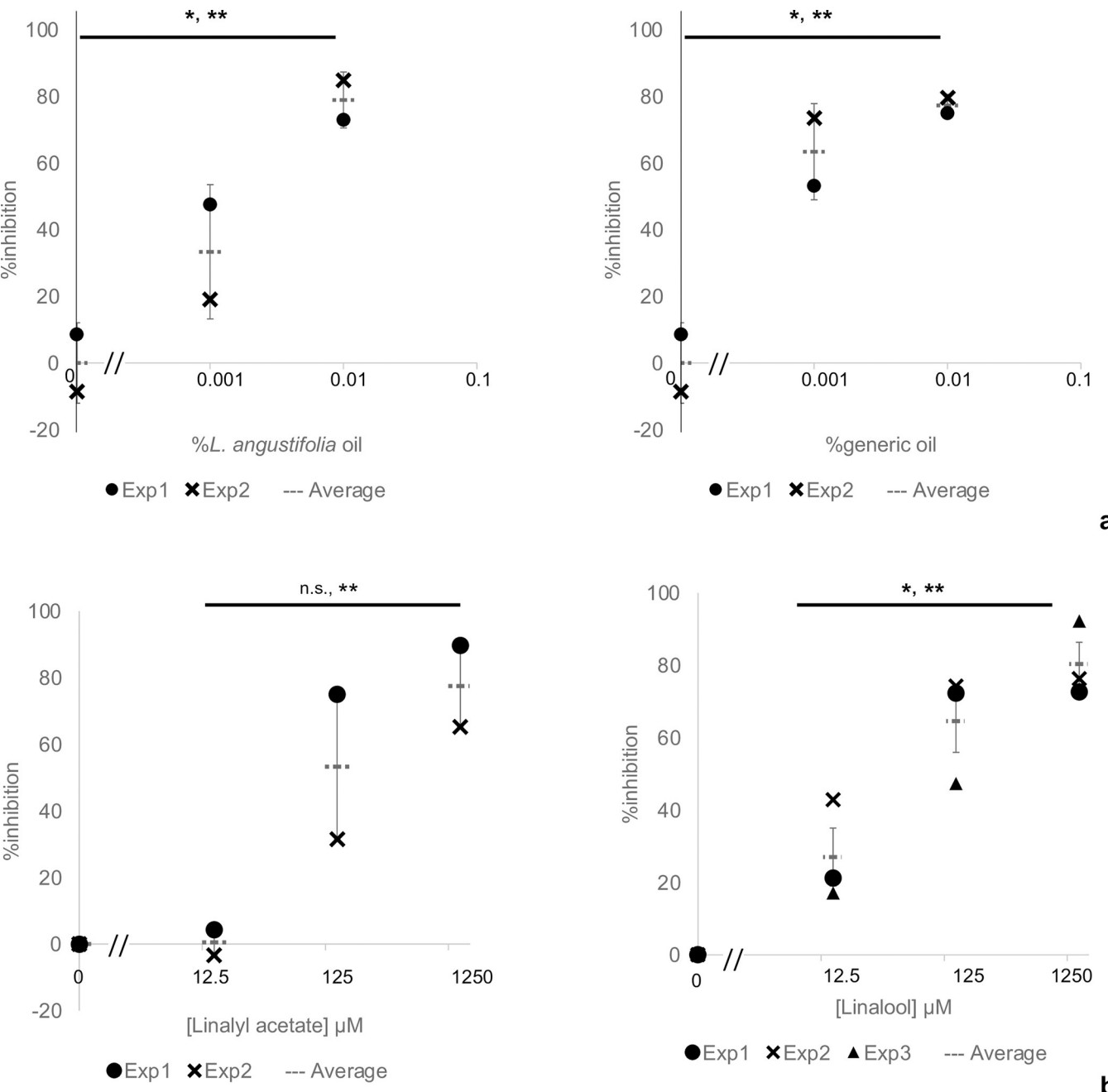

**Fig 2. *L. angustifolia* essential oil and its major components inhibit the induced expression of neurotrophic factor Artemin.** HaCaT cells were incubated in culture medium containing 5 μM FICZ with or without the indicated amounts of agents for 24 h. Total RNA was then extracted and reverse transcribed to cDNA that was subjected to qPCR using the primer set listed in S2 Table in S1 File. Artemin expression was normalized to GAPDH expression. Percentage inhibition was then calculated with FICZ alone as 0%, indicating no inhibition, and without FICZ as 100% indicating complete inhibition. The dot markers in the graphs represent data from two or three independent experiments. The vertical and horizontal bars represent the standard deviation and the mean, respectively. Comparisons of (a) *L. angustifolia* essential oil and generic lavender oil, and (b) linalyl acetate and linalool for inhibition of induced Artemin expression are shown. The statistical significance of the data points among the different concentrations and the dose-dependency were determined by the Kruskal-Wallis test or one-way ANOVA, and the Williams test, respectively. *, **$P<0.05$ (n = 3).

0.0014% ± 0.00003%, and $CV_{70}$ values of 0.032% ± 0.0034% and >0.04%, respectively. Although no positive criteria have been defined for mixtures such as essential oils, these can be weakly positive in this test if the positive criterion is cell viability of >70% at $EC_{1.5}$. Similarly,

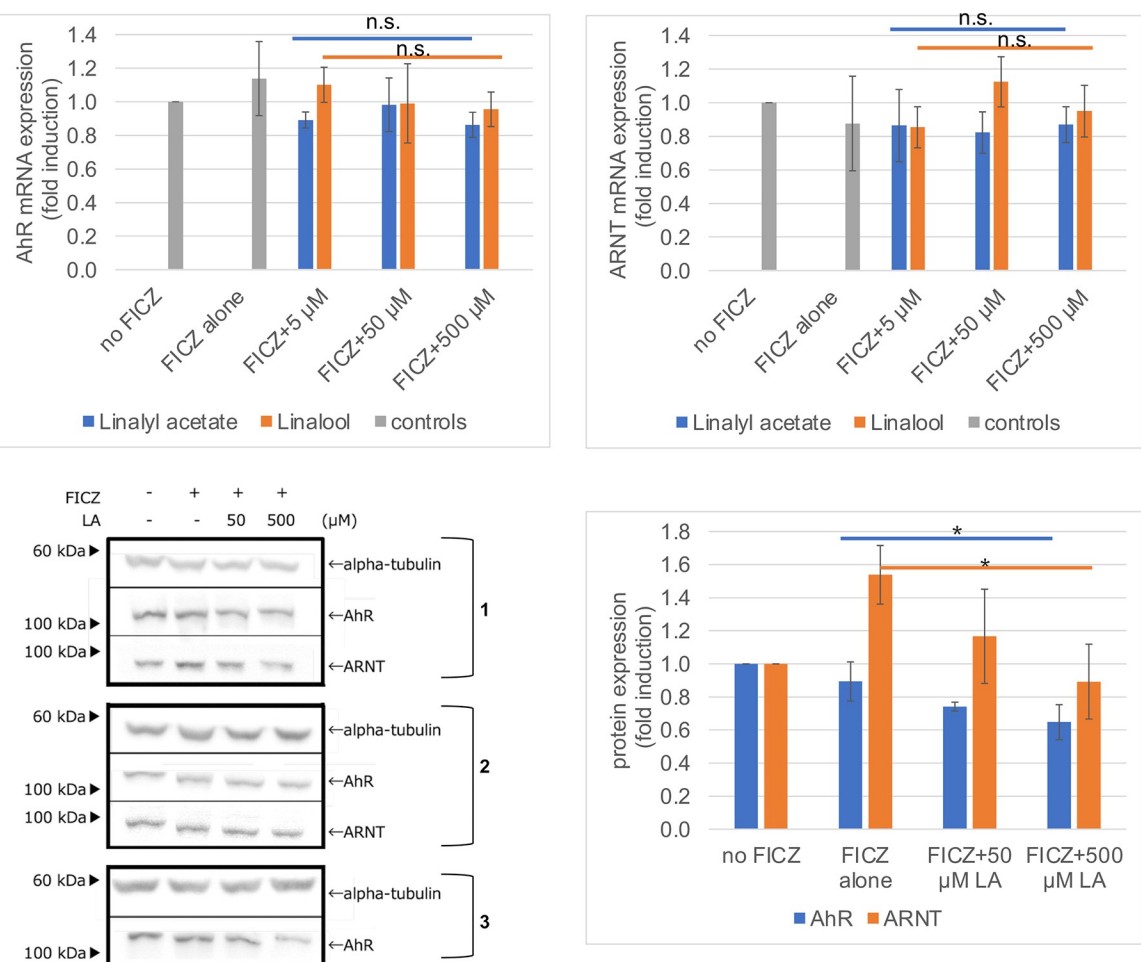

**Fig 3. The major component of *L. angustifolia* essential oil, linalyl acetate, promotes AhR and ARNT protein degradation and not their mRNA downregulation.** a, b) HaCaT cells incubated in culture medium containing the indicated agents for 24 h. Total RNA was then extracted and reverse transcribed to cDNA that was subjected to qPCR using the primer set listed in S2 Table in S1 File. AhR and ARNT expression was normalized to GAPDH expression. Fold induction was calculated as the vehicle set at 1. Values in the graphs represent the mean ± SD of two independent experiments. There were no statistically significant differences under all conditions. c, d) HaCaT cells were incubated in culture medium containing the indicated agents for 24 h. c) Cells were lysed and subjected to western blotting with antibodies against AhR, ARNT, and α-tubulin. All the Western blots are shown from two (anti-ARNT) or three (anti-AhR and anti-gamma tubulin) independent experiments. The right-side numbers are experimental numbers. d) Band intensity of each lane was quantified by ImageJ and normalized to α-tubulin. The intensities are relative to the vehicle set at 1. Values in the graphs represent the mean ± SD of two or three independent experiments. *$p < 0.05$, compared with FICZ alone. LA, linalyl acetate.

linalyl acetate, THL, and DHL were weakly positive in this test, whereas linalool was negative, which was consistent with a previous report [35].

## *L. angustifolia* essential oil and its major components are negative in the dendritic cell activation assay for skin sensitization evaluation

To confirm the skin-sensitizing potential of *L. angustifolia* essential oil and linalyl acetate, we performed the dendritic cell activation assay, a test to evaluate the third key event of the skin sensitization reaction using human monocyte-like cell line U937. U937 cells were incubated with test substances for 48 h and then subjected to flow cytometry with anti-CD86 antibody and PI staining. The $EC_{150}$ and $CV_{70}$ of each test substance are shown in Table 2. The negative

**Table 1. $EC_{1.5}$ and $CV_{70}$ in the keratinocyte activation assay.**

| agents | $EC_{1.5}$ | $CV_{70}$ |
|---|---|---|
| *L. angustifolia* essential oil | 0.0072±0.00065% | 0.032±0.0034% |
| Generic lavender oil | 0.0014±0.00003% | >0.04% |
| Linalyl acetate | 222.1±77.6 μM | 1466±45.4 μM |
| Linalool | - | >2000 μM |
| THL | 107.8±41.4 μM | 1416±90.6 μM |
| DHL | 340.0±166.8 μM | 1319±52.3 μM |
| Ethylene glycol dimethacrylate* | 4.1±0.52 μM | 523±59.0 μM |

Values in the Table represent the mean ± SD of three independent experiments.

*known keratinocyte activator EGDMA

$EC_{1.5}$ and $CV_{70}$ were calculated by applying the raw data shown in S1a, S1b Fig in S1 File to equation 1 in 442D [27].

criterion for this assay is defined in OECD TG442E [30], i.e., the S.I. value of CD86 is less than 150% and cell viability is greater than 70% and when two independent tests are both positive or negative, they are considered positive and negative, respectively. Trinitrobenzenesulfonic acid and lactic acid were used as positive and negative controls, respectively. Raw data to calculate the S.I., $CV_{70}$ and $EC_{150}$ are shown in S4 Table in S1 File. As shown in Table 2, because the S.I. values of *L. angustifolia* essential oil and linalyl acetate's were less than 150 at their $CV_{70}$ values, these agents are considered negative in this test. Conversely, the $EC_{150}$ values of generic lavender oil were 0.029% and 0.019% in two independent tests, whereas their $CV_{70}$ values were greater than 0.04%. These values met the positive criteria, indicating that generic lavender oil was positive in this test. Linalool has also been shown to be positive in this assay [35].

## Overall skin sensitization potential of *L. angustifolia* essential oil and its major components

Skin sensitization is a type of allergic reaction that occurs upon contact with a chemical substance. Skin sensitization reactions are summarized in the form of an AOP consisting of four key events. The first key event is covalent bond formation, the second is keratinocyte activation, the third is dendritic cell activation, and the fourth is T cell activation. Each event except the fourth evaluated by *in vitro* tests was partly assessed in this study. The overall skin sensitization potential of *L. angustifolia* essential oil, linalyl acetate, linalool, and generic lavender oil are summarized in Table 3. Because each test may have false positive and negative results, the skin-sensitizing potential of the test substance was comprehensively determined by multiple tests. By considering all the data, *L. angustifolia* essential oil and linalyl acetate had very weak or negative skin-sensitization potentials, but had strong AD inhibitory effects.

**Table 2. $EC_{150}$ and $CV_{70}$ in the dendritic cell activation assay.**

| agents | $EC_{150}$ | $CV_{70}$ |
|---|---|---|
| *L. angustifolia* essential oil | - | 0.021±0.038% |
| Generic lavender oil | 0.024±0.0053% | >0.04% |
| Linalyl acetate | - | 16±3.8 μg/mL |

Values in the Table represent the mean ± SD of two independent experiments.

$EC_{150}$ and $CV_{70}$ were calculated by applying the raw data shown in S4 Table in S1 File to equations 1 and 2 in the Materials and Methods.

## Discussion

We investigated whether *L. angustifolia* essential oil and its major components, linalyl acetate and linalool, have an AD inhibitory effect using an AD cell model, which had an AD inhibitory effect by inhibiting AhR activation (Fig 1). Furthermore, they decreased induced Artemin expression (Fig 2), confirming their AD inhibitory effect in combination with the results from the AD cell model. The protein, but not mRNA, levels of AhR and ARNT were decreased by linalyl acetate (Fig 3), suggesting that linalyl acetate contributes to protein degradation of AhR and/or ARNT. Finally, to evaluate the safety of *L. angustifolia* essential oil and its major components on skin, skin sensitization tests, such as keratinocyte and dendritic cell activation assays, were conducted (Tables 1 and 2). The results including ours and others' observations are summarized in Table 3. Collectively, this study shows that *L. angustifolia* essential oil and linalyl acetate barely have a skin-sensitization potential, but have a strong AD inhibitory effect.

Our results indicate that linalyl acetate in *L. angustifolia* essential oil is a major contributor to AD inhibitory effects and lavender oils with high linalyl acetate contents would be effective for AD treatment, although other components in *L. angustifolia* essential oil, which we did not test, might be other contributors. We showed that FICZ-induced Artemin expression was decreased by *L. angustifolia* essential oil and linalyl acetate. This can be confirmed by adding *L. angustifolia* essential oil- or linalyl acetate-treated keratinocyte culture supernatants to human neuroblastoma cell line SY-SY5Y and measuring their proliferation [36]. Another important factor, in addition to Artemin, involved in the pathogenesis of atopic dermatitis is thymic stromal lymphopoietin (TSLP), which is known to activate dendritic cells and promote Th2-type immune responses [37]. While TSLP has been extensively studied as a potential therapeutic target for atopic dermatitis, our model cell line does not express TSLP. Therefore, we were not able to examine the effects on TSLP expression in our study.

We concluded that the molecular mechanism underlying the AD inhibitory effect of *L. angustifolia* essential oil and linalyl acetate was promotion of AhR and ARNT protein degradation (Fig 3B) because *L. angustifolia* essential oil and linalyl acetate did not affect nuclear translocation or the mRNA levels of AhR and ARNT (Fig 3A, respectively). AhR acts as an E3

**Table 3. Four key events in the skin sensitization reaction and reactivities of *L. angustifolia* essential oil, generic lavender oil, linalyl acetate, and linalool.**

| Key event | 1 | 2 | 3 | 4 | all |
|---|---|---|---|---|---|
| test | Amino acid derivative reactivity assay | Keratinocyte activation assay | Dendritic cell activation assay | Local lymph node assay | Patch test (human) |
| *L. angustifolia* essential oil | N.D. | ±* | -* | N.D. | ±\$ \$ \$ |
| Generic lavender oil | N.D. | ±* | +* | N.D. | N.D. |
| Linalyl acetate | N.D. | ±* | -* | ±\$ | ±\$ \$ |
| Linalool | -** | -*, *** | +*** | +***,# | 0.2% positive##, ### |

N.D., not determined

*determined in this study

**[48]

***[35]

#[45]

##[49]

###[50]

\$[44]

\$ \$[44]

\$ \$ \$[51]

ubiquitin ligase under certain conditions [38] and AhR is degraded by ubiquitin proteasomes [39]. Therefore, *L. angustifolia* essential oil and linalyl acetate may contribute to degrading AhR and ARNT by promoting the ubiquitin proteasomal system. Future experiments with inhibitors of nascent protein synthesis and proteasome inhibitors would confirm this conjecture. However, because AhR is regulated by the antioxidative response [40], the mechanism underlying the AD inhibitory effect might not be this simple.

The keratinocyte activation assay, known as the Keratinosens™ skin sensitization assay, which evaluates the second key event in AOP of skin sensitization, showed that linalyl acetate, *L. angustifolia* essential oil, generic lavender oil, THL, and DHL were weakly positive, and linalool was negative. The $EC_{1.5}$ of *L. angustifolia* essential oil used here was 0.0072%. Because the percentage weight of linalyl acetate in this essential oil is 43%, the estimated $EC_{1.5}$ of linalyl acetate can be calculated to be 141 μM. However, the actual $EC_{1.5}$ of linalyl acetate was 222 μM, suggesting that this oil might contain other components in addition to linalyl acetate, which cause the positivity. Other components include terpenes and other alcohols although they were less than 1/10~1/100 of linalyl acetate. Importantly, not all positive compounds in this assay are skin sensitizers. For example, resveratrol and L-ascorbic acid, both used in cosmetics, are positive in this assay, but are not skin sensitizers [41]. Compounds that indirectly exert antioxidant effects by inducing expression of antioxidant genes are thought to be possible false positives in this assay [42]. *L. angustifolia* essential oil and linalyl acetate also have antioxidant properties [12, 42]. Therefore, it is possible that the positivity in this study was due to false positives in the ARE reporter assay to evaluate skin sensitization.

Because oxidation of linalool and linalyl acetate results in hydroperoxides, which increase skin sensitization [43, 44], a less oxidizable compound is desirable for practical use as an AD suppressant. Linalool has two unsaturated double bonds at two different locations, which can be easily oxidized [31]. We tested 6,7-dihydrolinalool (DHL) and tetrahydrolinalool (THL), hydrogenated forms of linalool at one and two unsaturated double bond(s), respectively. Both were weakly positive in the ARE reporter assay, although unfortunately, they exhibited low activity as an AD suppressant. Linalyl acetate also has such unsaturated double bonds and it would be interesting to test the hydrogenated forms of linalyl acetate for skin sensitization and AD suppression.

Another *in vitro* skin sensitization assay addressing the third key event, i.e., activation of dendritic cells of the AOP for skin sensitization, is the U-SENS™ assay that was performed in accordance with OECD TG442E [30]. In this assay, *L. angustifolia* essential oil and linalyl acetate were negative, whereas generic lavender oil was positive. Linalool is positive in this assay and another assay that evaluates the same key event [35, 45]. Chemical contents in the generic lavender oil used in this study are unknown. However, it may contain more components that may cause positivity in this assay, such as linalool.

The latest OECD guideline indicates that skin sensitization can be assessed by assays that evaluate the first to third key events in the AOP of skin sensitization or the first and third, and by the quantitative structure–activity relationship, an *in silico* method to predict chemicals with the potential to cause adverse effects on the basis of their chemical structures [46]. Although we did not perform either quantitative structure–activity relationship or the assay that evaluates the first key event, as summarized in Table 3, the overall prediction of linalyl acetate for skin sensitization would be weak or almost none on the basis of the positivity for the second key event, negativity for the third key event, weak positivity for the LLNA [44], and negativity in a patch test [47]. Similarly, *L. angustifolia* essential oil can also be predicted to cause weak or almost no skin sensitization based on the OECD guideline. However, generic lavender oil was positive for both the second and third events, and is likely to be a skin sensitizer.

## Conclusions

*L. angustifolia* essential oil and its major components, linalool and linalyl acetate, have AD inhibitory effects and almost no skin sensitization shown by this study, suggesting that *L. angustifolia* essential oil can be useful as an AD suppressant. The results presented here may be limited due to the oils and assays used in this study. Importantly, similar to several other agents, their efficacy is highly dependent on the optimal concentration range. Therefore, we respectfully suggest that their use at higher concentrations may not result in the desired therapeutic outcomes.

## Supporting information

**S1 Raw images. It contains the original blot images with their respective navigations and files for the statistical analyses.** The filenames are as follows: a. S1_raw_images_Figure3C.pdf b. 15.JPG c. 21.JPG d. 22.JPG e. 2201241741391.tif f. 2202041813141.tif g. 2202041817223.tif h. 2202041818381.tif i. 2202071847014.tif j. 2202071852122.tif k. 2202071853293.tif l. 2202071859374.tif m. S1_raw_images_FigureS2.pdf n. 7.JPG o. 8.JPG p. 2112201615449.tif q. 2112211700287.tif r. 2201132050057.tif s. Williams_test.txt t. grouped_scatter_plot_Bartlett_ANOVA_en.ipynb.
(ZIP)

**S1 File.**
(PDF)

## Acknowledgments

We thank the members of the Organelle Lab (Okayama University) for their assistance.

We thank Mitchell Arico from Edanz (https://jp.edanz.com/ac) for editing a draft of this manuscript.

## Author Contributions

**Conceptualization:** Haruna Sato, Kosuke Kato, Yoshimasa Nakamura, Yoshio Tsujino, Ayano Satoh.

**Data curation:** Haruna Sato, Kosuke Kato, Mayuko Koreishi.

**Formal analysis:** Haruna Sato.

**Funding acquisition:** Yoshimasa Nakamura, Yoshio Tsujino, Ayano Satoh.

**Methodology:** Haruna Sato, Kosuke Kato, Yoshimasa Nakamura.

**Resources:** Haruna Sato, Kosuke Kato, Mayuko Koreishi, Yoshio Tsujino.

**Supervision:** Ayano Satoh.

**Validation:** Haruna Sato, Kosuke Kato.

**Visualization:** Haruna Sato.

**Writing – original draft:** Haruna Sato.

**Writing – review & editing:** Yoshimasa Nakamura, Yoshio Tsujino, Ayano Satoh.

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
