## [Decision Letter · Decision Letter 0]

7 Jun 2023

PONE-D-23-13536Aromatic oil from lavender as an atopic dermatitis suppressantPLOS ONE

Dear Dr. Satoh,

Thank you for submitting your manuscript to PLOS ONE. After careful consideration, we feel that it has merit but does not fully meet PLOS ONE’s publication criteria as it currently stands. Therefore, we invite you to submit a revised version of the manuscript that addresses the points raised during the review process.

We look forward to receiving your revised manuscript.

Kind regards,

Mozaniel Santana de Oliveira, Ph.D

Academic Editor

PLOS ONE

Journal Requirements:

3. Please expand the acronym “JSPS” (as indicated in your financial disclosure) so that it states the name of your funders in full.

Additional Editor Comments:

Dear Dr. Satoh,

I received the reviews on your manuscript, the reviewers requested that more corrections be incorporated throughout the work.

Changes must be highlighted in yellow so that we can track them.

Reviewers' comments:

Reviewer's Responses to Questions

**Comments to the Author**

1. Is the manuscript technically sound, and do the data support the conclusions?

Reviewer #1: Yes

Reviewer #2: Yes

2. Has the statistical analysis been performed appropriately and rigorously? 

Reviewer #1: Yes

Reviewer #2: Yes

3. Have the authors made all data underlying the findings in their manuscript fully available?

Reviewer #1: Yes

Reviewer #2: Yes

4. Is the manuscript presented in an intelligible fashion and written in standard English?

Reviewer #1: Yes

Reviewer #2: Yes

5. Review Comments to the Author

Reviewer #1: Authors established a human keratinocyte cell line stably expressing a NanoLuc reporter gene downstream of XRE. They found that 6-formylindolo[3,2-b]carbazole (FICZ), a tryptophan metabolite and known inducer of the XRE, increased reporter and Artemin mRNA expression, indicating that FICZ-treated cells could be a model for AD. Lavender essential oil has been used in folk medicine to treat AD, but the scientific basis for its use is unclear.

In the present study, they investigated the efficacy of lavender essential oil and its major components, linalyl acetate and linalool, to suppress AD and sensitize skin using the established AD model cell line, and keratinocyte and dendritic cell activation assays. Their results indicated that lavender essential oil from L. angustifolia and linalyl acetate exerted a strong AD inhibitory effect and almost no skin sensitization. Their model is useful in that it can circumvent the practice of using animal studies to evaluate AD medicines.

# Authors must go through very recent papers on how essential oils can find their applications in treating disorders, etc. Here are few references to cite. DOI: 10.3390/antiox11122410; DOI: 10.1016/j.jics.2021.100088, DOI: 10.1016/j.tifs.2022.10.012.

# Rest is fine and I recommend manuscript acceptance following the minor suggestions.

Reviewer #2: The manuscript investigated possible molecular mechanisms responsible for the protective effect of lavender essential oils in atopic dermatitis, using in vitro models. The authors identified the promotion of degradation of AhR and ARNT proteins as main molecular mechanisms responsible for the effect. The study is interesting adding new insights to the possible use of natural products in the treatment of atopic dermatitis, however a few minor corrections (clarifications) are needed ahead of publishing:

1. In the Introduction (Lines 46-47) the authors state that "It affects 15%–20% of children and 1%–3% of adults worldwide

and patient numbers are increasing''- please provide the reference number for this epidemiological information.

2. In the same Introduction (Lines 65-66) the authors state that "Currently, steroids and immunosuppressive agents are used to treat AD. However, they are only symptomatic treatments''. The authors should know that new drugs for AD have been recently authorized by FDA and EMA: dupilumab, baricitinb and tralokinumab (anti IL-13 agent) which can modulate specific pathogenetic mechanisms of AD. It is more appropriate to say that despite the recent developments in the pharmacotherapy of AD, there is still a real need for the discovery of new molecules capable of better controlling AD.

3. In the Results section the authors state that ''Both L. angustifolia essential oil and generic lavender oil inhibited AD". Please bear in mind that your study evaluated specific molecular events and not the extent of the real disease which can be evaluated in vivo (extent and progression of the lesions, etc). In fact the tested essential oils inhibited AhR activation not the disease itself which is a very complex entity.

6. PLOS authors have the option to publish the peer review history of their article (what does this mean?). If published, this will include your full peer review and any attached files.

Reviewer #1: **Yes: **No

Reviewer #2: No

---

## [Author Response · Author response to Decision Letter 0]

15 Nov 2023

Reviewers' comments, Our answers

Reviewer #1: 

# Authors must go through very recent papers on how essential oils can find their applications in treating disorders, etc. Here are few references to cite. DOI: 10.3390/antiox11122410; DOI: 10.1016/j.jics.2021.100088, DOI: 10.1016/j.tifs.2022.10.012.

 We are grateful for the enlightening insights shared in the recent papers. In the revised version of our manuscript, we have expanded our Introduction section in the context of the new applications of EOs by citing these papers.

# Rest is fine and I recommend manuscript acceptance following the minor suggestions.

We are very grateful for the reviewer's positive evaluation of our manuscript and recommendation for acceptance.

Reviewer #2: 

1. In the Introduction (Lines 46-47) the authors state that "It affects 15%–20% of children and 1%–3% of adults worldwide and patient numbers are increasing''- please provide the reference number for this epidemiological information.

 We apologize for the inadvertent omission of the corresponding reference in our original submission and appreciate your diligence in identifying this oversight. We have now properly included the appropriate citation [1] in our revised manuscript (https://doi.org/10.1159/000370220). 

2. In the same Introduction (Lines 65-66) the authors state that "Currently, steroids and immunosuppressive agents are used to treat AD. However, they are only symptomatic treatments''. The authors should know that new drugs for AD have been recently authorized by FDA and EMA: dupilumab, baricitinb and tralokinumab (anti IL-13 agent) which can modulate specific pathogenetic mechanisms of AD. It is more appropriate to say that despite the recent developments in the pharmacotherapy of AD, there is still a real need for the discovery of new molecules capable of better controlling AD.

We appreciate your critical feedback. In response to your insightful suggestions, we have included the appropriate descriptions in the Introduction section of our revised manuscript.

3. In the Results section the authors state that ''Both L. angustifolia essential oil and generic lavender oil inhibited AD". Please bear in mind that your study evaluated specific molecular events and not the extent of the real disease which can be evaluated in vivo (extent and progression of the lesions, etc). In fact the tested essential oils inhibited AhR activation not the disease itself which is a very complex entity.

 We have revised the Results section to more accurately reflect that our study observed inhibition of AhR activation by both L. angustifolia essential oil and generic lavender oil, rather than direct inhibition of AD.

---

## [Decision Letter · Decision Letter 1]

13 Dec 2023

Aromatic oil from lavender as an atopic dermatitis suppressant

PONE-D-23-13536R1

Dear Dr. Satoh,

We’re pleased to inform you that your manuscript has been judged scientifically suitable for publication and will be formally accepted for publication once it meets all outstanding technical requirements.

Kind regards,

Mozaniel Santana de Oliveira, Ph.D

Academic Editor

PLOS ONE

Additional Editor Comments (optional):

Reviewers' comments:

Reviewer's Responses to Questions

**Comments to the Author**

1. If the authors have adequately addressed your comments raised in a previous round of review and you feel that this manuscript is now acceptable for publication, you may indicate that here to bypass the “Comments to the Author” section, enter your conflict of interest statement in the “Confidential to Editor” section, and submit your "Accept" recommendation.

Reviewer #1: All comments have been addressed

Reviewer #2: All comments have been addressed

2. Is the manuscript technically sound, and do the data support the conclusions?

Reviewer #1: Yes

Reviewer #2: Yes

3. Has the statistical analysis been performed appropriately and rigorously? 

Reviewer #1: Yes

Reviewer #2: Yes

4. Have the authors made all data underlying the findings in their manuscript fully available?

Reviewer #1: Yes

Reviewer #2: Yes

5. Is the manuscript presented in an intelligible fashion and written in standard English?

Reviewer #1: Yes

Reviewer #2: Yes

6. Review Comments to the Author

Reviewer #1: Authors have addressed all comments. The manuscript can be accepted. All queries are now properly resolved

Reviewer #2: I am glad that the authors have made the necessary corrections , therefore I recommend the publication of the revised manuscript.

7. PLOS authors have the option to publish the peer review history of their article (what does this mean?). If published, this will include your full peer review and any attached files.

Reviewer #1: No

Reviewer #2: No

---

## [Editor Report · Acceptance letter]

27 Dec 2023

PONE-D-23-13536R1 

PLOS ONE

Dear Dr. Satoh, 

I'm pleased to inform you that your manuscript has been deemed suitable for publication in PLOS ONE. Congratulations! Your manuscript is now being handed over to our production team.

Kind regards, 

on behalf of

Dr. Mozaniel Santana de Oliveira 

Academic Editor

PLOS ONE